Novel Systems Biology Techniques

# The Signal and the Noise: Characteristics of Antisense RNA in Complex Microbial Communities

Thomas Yssing Michaelsen,[a] Jakob Brandt,[a] Caitlin Margaret Singleton,[a] Rasmus Hansen Kirkegaard,[a] Johanna Wiesinger,[b] Nicola Segata,[c] Mads Albertsen[a]

[a]Center for Microbial Communities, Aalborg University, Aalborg, Denmark
[b]Centre for Microbiology and Environmental Systems Science, University of Vienna, Vienna, Austria
[c]Department CIBIO, University of Trento, Trento, Italy

**ABSTRACT** High-throughput sequencing has allowed unprecedented insight into the composition and function of complex microbial communities. With metatranscriptomics, it is possible to interrogate the transcriptomes of multiple organisms simultaneously to get an overview of the gene expression of the entire community. Studies have successfully used metatranscriptomics to identify and describe relationships between gene expression levels and community characteristics. However, metatranscriptomic data sets contain a rich suite of additional information that is just beginning to be explored. Here, we focus on antisense expression in metatranscriptomics, discuss the different computational strategies for handling it, and highlight the strengths but also potentially detrimental effects on downstream analysis and interpretation. We also analyzed the antisense transcriptomes of multiple genomes and metagenome-assembled genomes (MAGs) from five different data sets and found high variability in the levels of antisense transcription for individual species, which were consistent across samples. Importantly, we challenged the conceptual framework that antisense transcription is primarily the product of transcriptional noise and found mixed support, suggesting that the total observed antisense RNA in complex communities arises from the combined effect of unknown biological and technical factors. Antisense transcription can be highly informative, including technical details about data quality and novel insight into the biology of complex microbial communities.

**IMPORTANCE** This study systematically evaluated the global patterns of microbial antisense expression across various environments and provides a bird's-eye view of general patterns observed across data sets, which can provide guidelines in our understanding of antisense expression as well as interpretation of metatranscriptomic data in general. This analysis highlights that in some environments, antisense expression from microbial communities can dominate over regular gene expression. We explored some potential drivers of antisense transcription, but more importantly, this study serves as a starting point, highlighting topics for future research and providing guidelines to include antisense expression in generic bioinformatic pipelines for metatranscriptomic data.

**KEYWORDS** RNAseq, antisense RNA, *cis*-antisense RNA, meta-analysis, metatranscriptomics, review

This article followed an open peer review process. The review history can be read here.

Address correspondence to Thomas Yssing Michaelsen, tym@bio.aau.dk.

A much-needed meta-analysis of antisense RNA in metatranscriptomes by @TYMichaelsen and co-authors @JakobBrandt90 @CaitySing @kirk3gaard @nsegata @MadsAlbertsen85

Transcriptomics is an established tool to quantify bacterial RNA expression and regulation on a whole-organism level (1), which increasingly also includes a wide repertoire of small regulatory noncoding RNAs (sRNAs) (2, 3). This field received a dramatic boost with the introduction of next-generation RNA sequencing (RNAseq), a technique initially applied mostly on human samples but which has also allowed

unprecedented insight into the overall characteristics of complete microbial transcriptomes (1). The probe-independent nature of RNAseq provides a supposedly unbiased means of identifying novel putative genes and sRNA elements that may be missing from the genomic annotation (2). In particular, strand-specific RNAseq enables one to retain the directionality of the native RNA segment when sequenced and to quantify both mRNA and antisense RNA (asRNA), here defined as the sRNA matching the template DNA strand inside a gene (4), which may act as a biological regulator of mRNA and transcription (2). An example of this is the occurrence of spurious promoters (5). As bacterial promoters are characterized by low information content, these could arise by random mutations throughout the genome (6) and cause transcription at constant rates across the genome without biological function, including asRNAs (5, 7).

Multiple studies have identified asRNA to be ubiquitous in bacteria (8, 9) and increasingly relevant in archaea (10). These studies identified asRNA primarily through genome-wide searches for putative sRNA and transcriptome analysis of single-culture bacteria (11). Across different bacteria, the number of genes with asRNA varies substantially, from more than 80% in *Pseudomonas aeruginosa* PA14 (12) to 2.3% in *Lactococcus lactis* (13). The functionality of asRNA has been extensively studied, and several interactions between specific asRNAs and mRNAs have been described (2), including the ability to inhibit the expression of adjacent mRNA (9). In *Staphylococcus aureus*, it is suspected that at least 75% of mRNAs are modulated by asRNA (14), and a substantial number of asRNA-mRNA interactions were directly identified in *Escherichia coli* (15) and *Salmonella* (16). In contrast, studies comparing the conservation and expression of promoters on the antisense strand across species and strains within a species did not find good correlations, suggesting that antisense RNA is nonfunctional and a by-product of the transcriptional machinery (17, 18).

Pure-culture studies are powerful tools to quantify asRNA and identify specific asRNA-mRNA interactions but may not generalize to how the organism responds in complex microbial communities and its natural complex environment. The interplay between organisms changes their transcriptional pattern through responses to, e.g., competition or commensal interactions (19), but how this extends to asRNA remains largely unknown (20). Detailed studies of the presence and dynamics of asRNA in complex microbial communities are scarce; to our knowledge, only one study investigated asRNA in human gut communities (4). Consequently, our understanding of the variability and persistence of asRNA in complex communities, between and within different environments, is limited.

The onset of metatranscriptomics enabled the transcriptomes of multiple organisms to be examined simultaneously, providing a complete overview of the gene expression of the entire community without the need for culturing. This has made metatranscriptomics a powerful tool to detect and describe relationships between gene expression levels and community characteristics (20). When coupled with metagenomics for the *de novo* construction of metagenome-assembled genomes (MAGs), this provides a high-throughput culture-independent method for performing species-level analysis *in situ* (21), not limited to genomes from already well-characterized species (22). As increasing amounts of metatranscriptomic data are being generated by strand-specific RNA sequencing, we encourage researchers to embrace this extra dimension of the data adequately. In this study, we reviewed the different approaches for processing asRNA and highlight some of the emergent quantitative properties of asRNA, to provide a bird's-eye view of antisense transcription in complex microbial communities.

## RESULTS AND DISCUSSION

**Three approaches to handling asRNA in the literature.** We identified three common approaches to handling strand-specific metatranscriptomic RNAseq data (Table 1), which rely on substantially different assumptions about the origin of asRNA and its effect on study outcome. First, some studies ignore the strand-specific information of the data completely (23–26), implying that the expression of a given region will be the sum of sense and antisense expression. This simplifies downstream analysis

**TABLE 1** Relevant papers and their procedure for handling antisense RNA

| Reference | Environment | Method to determine strandness | Handling |
|---|---|---|---|
| 32 | Soil | ScriptSeq complete kit (bacteria) | Incorporated |
| 4 | Human gut | Custom protocol[a] | Incorporated |
| 28 | Human gut | NEBNext Ultra directional RNA kit | Filtered |
| 23 | Water | TruSeq stranded kit | Ignored |
| 24 | Human gut | RNAtag-seq[b] | Ignored |
| 26 | Human gut | ScriptSeq complete cold kit | Ignored |
| 62 | Soil | TruSeq stranded kit | Ignored |
| 25 | Human gut | RNAtag-seq[b] | Ignored |

[a]See reference 63.
[b]See reference 64.

and has been the traditional approach given that stranded RNAseq, conserving the sense RNA transcripts, is relatively new (27). Importantly, it is also based on the assumption that asRNA is irrelevant for the study outcome, intentionally or not. Second, other studies filter away the antisense mapping reads (28), a step which is also integrated into many popular tools designed for quantifying gene expression, such as HTSeq (29), Cufflinks (30), and featureCounts (31). This ensures the quantification of mRNA only. Third and finally, there are studies that directly incorporate the strand-specific signal into the analysis using statistical tests comparing sense and antisense expressions within each region of interest (4, 32). This may be used to filter away genes with low sense relative to antisense expression levels (32) or to identify genes with significant antisense expression (4).

**Antisense RNA is concentrated on a few genes that are functionally reduced.** To elucidate asRNA in strand-specific metatranscriptomes, we performed an explorative meta-analysis of the quantitative, functional, and genome-specific properties of asRNA in five public data sets, covering an anaerobic digester, water, bog and fen permafrost-associated wetlands, and human gut. The percentages of asRNA in the data sets varied from 1.1% to 10.4% for the human gut and water communities (Fig. 1A). The anaerobic digester, bog, and fen samples had a substantially higher percentage of asRNA, ranging from 33.2 to 90.8%. Surprisingly, only 2 to 3 genes accounted for up to 90% of all asRNAs in these communities, while these numbers were 22% and 61% in the human gut and water communities, respectively (Fig. 1B). We attempted to manually annotate the top 3 most asRNA-expressing genes in all five communities (15 in total) using BLASTp (https://blast.ncbi.nlm.nih.gov/Blast.cgi) and obtained no hits fulfilling a query coverage of ≥25%, an identity of ≥70%, and an E value of ≤0.001 (33, 34). Conversely, the abundances of the three genes with the highest median mRNA expression levels relative to total mRNA were highly variable (Fig. 1B). We checked the coverage profiles of these genes and found no spurious spikes of read coverage caused by uneven primer affinity, making it unclear what causes the consistent high-level expression of asRNA. Generally, the cumulative percentages of mRNA follow a log-linear dependence on the number of genes for all five communities (Fig. 1B). By removing the 10 most asRNA-expressing genes from the analysis (see Fig. S1 in the supplemental material), the percentage of asRNA was drastically reduced in the anaerobic digester, bog, and fen samples, and the cumulative percentage profiles approached those of human gut and water samples.

By plotting the combined mRNA and asRNA expression against the percentage of asRNAs for each gene, we observed a characteristic U shape across all five communities (Fig. 1C). Substantial percentages of gene expression mRNA/asRNA pairs are exclusively mRNA or asRNA across all communities. As high expression levels of asRNA could be attributed to annotation errors (4), we searched all annotated protein sequences against the AntiFam database (35). We found no hits, and although it is not clear how representative the database is for complex communities, the consistency across communities may suggest biological relevance. The presence of asRNA might have regulatory properties, as high levels of asRNA could act to block gene expression (3, 4). Such

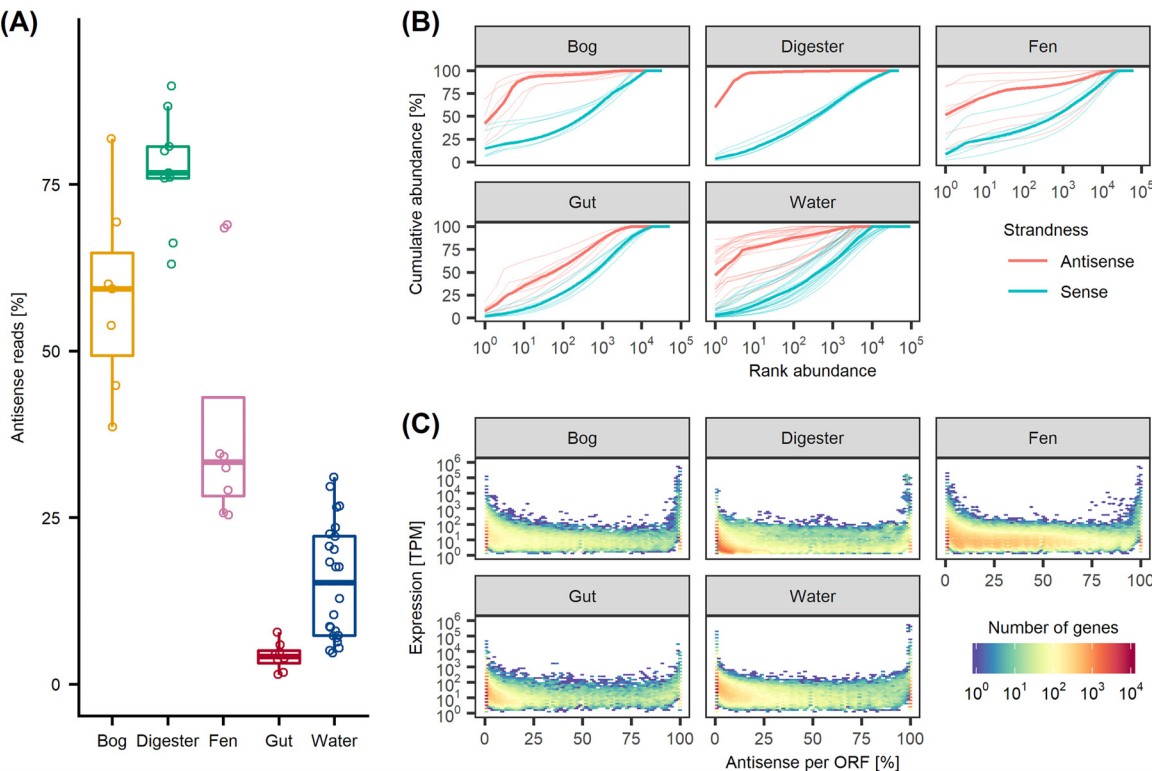

**FIG 1** Antisense transcription is a substantial amount of the data and dominated by few genes. (A) Percentage of sequenced reads which are antisense for each sample, grouped by community. (B) Rank-abundance curves for all samples within each community, colored by strandness. The x axis lists genes ranked in descending order according to relative abundance, while the y axis shows the cumulative abundance. Bold lines indicate the median for each colored group. (C) Relative amount of antisense expression for each observation (x axis) and total antisense and sense expression normalized to transcripts per million (TPM) (y axis). For results in this figure, sample-wide data sets were used (see Materials and Methods for details). ORF, open reading frame.

information could, for example, partition genes into suppressed, intermediate, and unsuppressed genes based on certain cutoff values of percent asRNA (4). Alternatively, the high percentages of genes with asRNA in bog and particularly fen samples have previously been attributed to DNA contamination, as double-stranded DNA during library preparation may separate, causing information on strandness to be lost (32). This approach discards large proportions of the data upfront and may be too conservative for many purposes. Furthermore, DNA contamination due to technical errors should occur randomly (8) and, therefore, evenly for all genes, canceling out any biases. We performed a simple experiment to test the effect of genomic DNA contamination on antisense RNA expression, simulated by a spike-in of DNA from the same sample prior to the removal of rRNA (Fig. S3). The percentage of antisense RNA reads per gene shifted upward as a function of gene DNA coverage in both the control and spiked-in samples, albeit the effect was substantially more profound in the spiked-in samples. As upward shifts were detected in both the control and spike-in samples, genomic DNA contamination is also likely to occur in many metatranscriptomes, readily detectable as an increase in asRNA expression.

To identify potential high-level biological drivers of asRNA expression, we investigated the enrichment of Clusters of Orthologous Groups (COG) functional categories in the five different environments. Antisense-enriched genes, which were defined as genes with asRNA accounting for ≥95% of the total RNA, were compared against the remaining background of genes (Fig. 2A). All COG categories except cytoskeleton (category Z) and nuclear structure (category Y) were detected in all communities (Fig. 2A). Between 44% and 67% of all genes with expression levels above the detection limit were assigned COG identifiers. The majority of COG categories were significantly reduced in asRNA-enriched genes across all five communities, with only "mobilome:

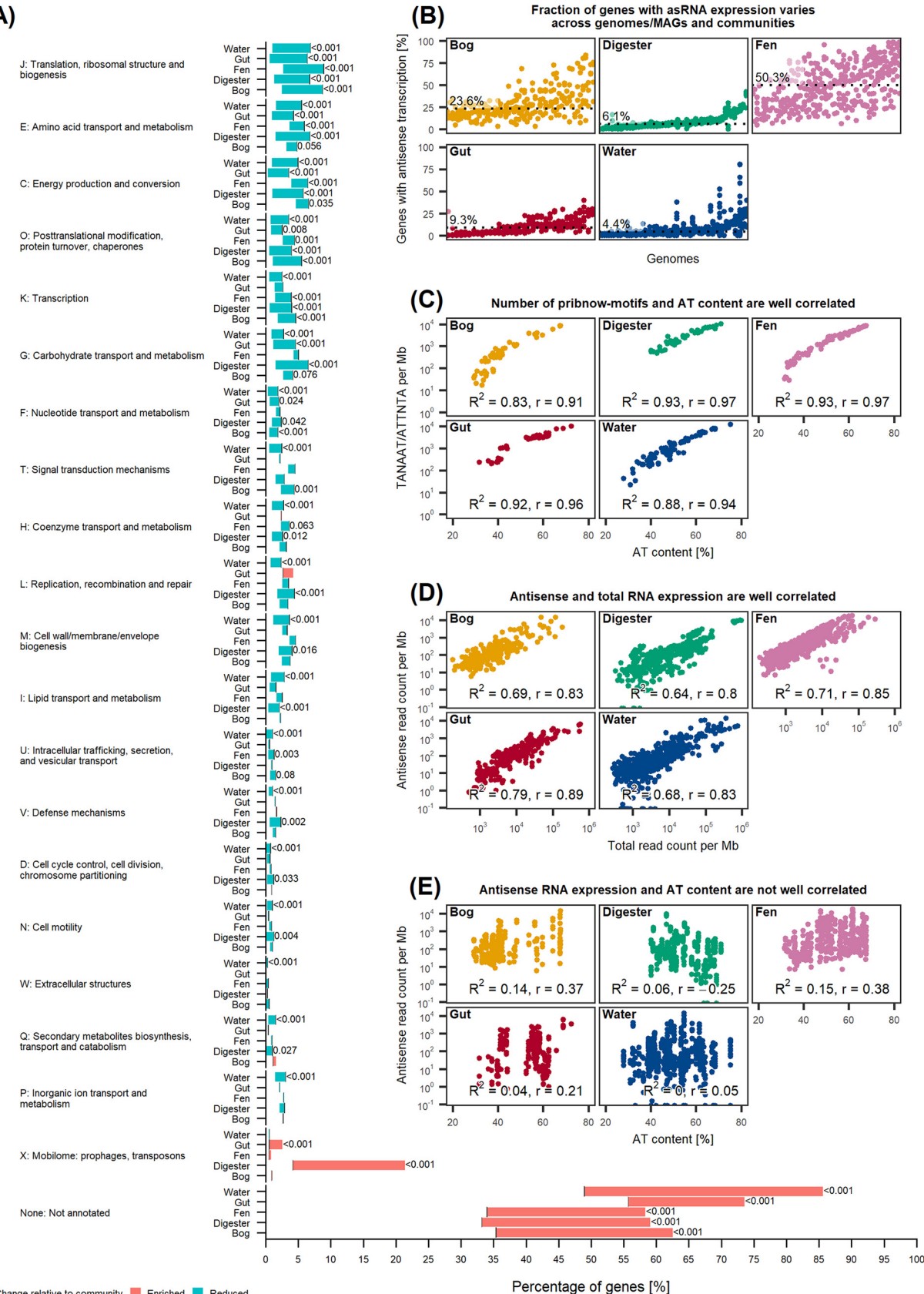

**FIG 2** Antisense transcription is not function specific and may be driven by several factors. (A) Listed from top to bottom is the enrichment/reduction for each COG functional category for each community, with the x axis showing the percentage of genes in either the COG functional

prophages, transposons" (category X) showing significant ($P < 0.001$) enrichment in the anaerobic and gut communities. However, genes without COG annotation were significantly enriched in all communities ($P < 0.001$), from means of 41% (33% to 56%) to 68% (58% to 86%) (Fig. 2A). This is in contrast to the findings of Bao and colleagues (4), who identified multiple COG categories to be enriched. However, a direct comparison with the results of this study is not possible, as we identified asRNA-enriched genes using a cutoff of the ratio of asRNA directly, in contrast to Bao et al. (4), who used a $P$ value cutoff from a statistical test. Regardless, this lack of functional assignment demands further research. Linking levels of asRNA to the suppression/activation of certain genes or pathways in a case-control setup might provide a path forward to integrate such information into metatranscriptomic studies (20).

**Levels of asRNA are genome specific and highly variable.** We also investigated the variability in antisense-expressing genes between genomes. To this end, we quantified the percentage of genes for each genome with significant asRNA expression (adjusted $P$ value of $<0.01$) using a binomial test (see Materials and Methods) (Fig. 2B). This varied substantially between communities, with averages from 4.4% to 50.3%. We further performed a univariate linear regression of all variables available that might influence the fraction of genes with significant antisense RNA expression (AT content, total RNA expression per megabase pair, sample- and genome-specific effects, and promoter motif occurrence per megabase pair). In all, except for the fen community, genome had the highest explanatory power, as quantified by the $R^2$ value (anaerobic digester, 90.8%; bog, 44.7%; fen, 36.51%; human gut, 55.3%; water, 42.9%), whereas sample-specific effects, representing sample-to-sample variability, were less profound (anaerobic digester, 0.8%; bog, 32.1%; fen, 45.9%; human gut, 10.8%; water, 9.2%). Total RNA expression had some explanatory power, with a mean $R^2$ of 22.1% (range, 12.2% to 26.6%), while AT content, promoter motif occurrence, and abundance (measured as genome coverage) all had $R^2$ values of $<10\%$ across all communities. We detected a marginally significant increase in $R^2$ values across communities when adding total RNA expression (one-tailed rank sum $P$ value of 0.08) or sample (one-tailed rank sum $P$ value of 0.08) to the linear model with genome as a regressor. We also performed the above-described analysis using a less conservative $P$ value of 0.05 for significant antisense RNA expression, to assess the robustness of the results (Fig. S2). Although we observed some shifts in the fraction of antisense RNA-expressing genes, the overall conclusion that genome alone explains most of the variability in antisense RNA expression remains the same. Genome-specific asRNA levels have been reported for pure cultures (5), and our study is the first to extend these findings beyond the human gut community (4). Thus, high intercommunity variability of genome-wide asRNA expression is a prevalent phenomenon. Such differences may have a profound effect when comparing gene expression levels between genomes and can bias the analysis if not properly accounted for. Interestingly, the extent of this may also be community specific, as genomes explained 36.5% to 90.8% of the variability in the percentages of genes with significant asRNA expression in water and anaerobic digester communities, respectively.

**FIG 2** Legend (Continued)

category or the entire community. Note the break between 10% and 50% on the *x* axis, where the group of nonannotated genes extends across the whole figure from left to right. The black vertical line at each horizontal bar indicates the percentage of annotated genes in the whole community, while bars indicate either percent reduction or enrichment in the subset of antisense-enriched genes, defined as genes with ≥95% of reads being antisense RNA. Only two-tailed adjusted *P* values of <0.2 are shown (see Materials and Methods for details). Shown are total numbers of genes and antisense-enriched subsets in digester (total, 48,889; subset, 936 [1.9%]), human gut (total, 53,059; subset, 927 [1.8%]), water (total, 95,695; subset, 5,339 [5.6%]), fen (total, 62,381; subset, 1,759 [2.8%]), and bog (total, 33,258; subset, 1,470 [4.4%]) samples. (B) For each community, genomes are plotted along the *x* axis, and the percentages of genes with significant (adjusted *P* value of <0.05) antisense expression within each genome are plotted on the *y* axis. Points located above the same genome represent the same genome across different samples. For each community, genomes are ordered along the *x* axis according to the median. Dotted lines and values indicate the median genome-wide percentages of genes with antisense transcription for each community. (C) Percent AT content for a given genome (*x* axis) and the number of Pribnow motifs in both directions of the sequence, normalized by genome length (*y* axis). (D) Number of sense RNA counts (*x* axis) and antisense RNA counts (*y* axis), normalized by genome length. (E) Percent AT content for a given genome (*x* axis) and number of antisense RNA read counts normalized by genome length (*y* axis). For this plot, genome-wide data sets were used (see Materials and Methods for details).

We attempted to extend the results of Lloréns-Rico et al. (5) to complex communities, stating that bulk asRNA is primarily the product of transcriptional noise arising from spurious promoters. We found a log-linear dependence of the number of Pribnow motifs per megabase pair on the AT content in all communities (Fig. 2C). This supports the findings that an increase in AT content leads to an increase in spurious promoter sites, consistent with simple combinatorics (5). Likewise, asRNA and total RNA expression per megabase pair were correlated (fig. 2D), but we did not find support for the positive effect of AT content on the expression of asRNAs per megabase pair (Fig. 2E). The lack of a relationship might be due to confounding of other variables, which, when accounted for, induce a positive relation between AT content and antisense RNA expression. We performed a Z-test using estimates and standard errors obtained from the models to test for a significant increase in the regression coefficient of AT content when adding a confounder to the model for each community. We found no effect after adjusting for total RNA expression (all $P > 0.05$) and sample (all $P > 0.3$). For genome, we found mixed results, with bog and water communities being significant ($P < 0.001$), which was not the case for the remaining communities ($P > 0.25$). For the digester and water communities, DNA sequencing data were available, which we used to compute the genomic DNA coverage of each genome. This had a limited explanatory power for asRNA expression (both $R^2 < 7.3\%$) and no confounding effect on AT content (all $P > 0.7$). Interestingly, we found DNA coverage to have explanatory power on a single-gene basis as discussed above for the DNA contamination experiment, suggesting that additional complexity is added at the genome level.

We have investigated two different parameterizations of asRNA expression, one with a significant fraction of asRNA per genome and one with the asRNA read count per megabase pair. These are complementary, with the former relying on a statistical test taking into account the expected level of asRNA and the measured level of mRNA, while the latter is assumption free and based exclusively on asRNA expression. The transcriptional machinery is complex, and such different approaches to analyze the data can highlight different aspects of antisense RNA. We did not observe a direct link between asRNA and pervasive transcription in the form of spurious promoter occurrence. This could be due to unknown molecular or biological processes that might mask this relation, or the promoters might not be actively transcribing (7), emphasizing the need for further studies to elucidate this.

**Role of asRNA in metatranscriptomics research.** It is not clear to what extent asRNA in metatranscriptomics reflects a biological rather than a stochastic process (5). Although a considerable body of literature is increasingly explaining the potential biological mechanisms behind asRNA, one should be careful generalizing these findings to metatranscriptomics. Reference genomes are often not available, meaning that genomes are constructed *de novo* from corresponding metagenomes (36). This often leads to incomplete annotation due to novelty, partially hampering downstream biological interpretation. Much of the current knowledge about asRNA is indeed based on pure cultures (20), which may not be representative of the intricate interactions that potentially occur in coexistence with other organisms in complex communities, where single cells within a population may be in different states of growth and dormancy simultaneously (37). High-throughput single-cell analysis of microbial communities has recently been developed (38) and would allow a single-cell perspective that might be needed to fully understand asRNA and mRNA interactions that are otherwise lost. Currently, there are limitations for the general application of single-cell analysis (38), and profiling of bulk microbial communities by high-throughput omics techniques remains a powerful tool to build metabolic networks and infer interactions between different organisms (21, 25, 39). We believe that asRNA expression could be integrated into such analyses, for example, to infer the suppression of genes based on asRNA/mRNA ratios, as additional information to the trivial reporting of up- and downregulation. From our literature search, we found three approaches to handling asRNA (Table 1), of which ignoring asRNA or quantifying only the sense RNA is by far the most

mSystems®

common. Ignoring strand specificity might be reasonable for the majority of genes, given that these do not express asRNA (Fig. 1C). However, metatranscriptomic studies are often explorative, in the sense of identifying biomarkers that can differentiate groups (40), and as genes with mainly asRNA tend to be highly expressed, these are likely to be emphasized during standard genome-wide association studies (GWASs). Such genes would be false positives or misinterpreted as being expressed in the traditional way (4). Quantifying only the directly transcribed RNA avoids such false positives and should be the naive, bare-minimum requirement for quantifying gene expression in metatranscriptomes. Of course, we believe that the optimum would be to include asRNA as a separate entity in analyses whenever possible, as important information may be retrieved. For example, high levels of asRNA could have inhibitory effects on downstream protein production (2, 3). This could be investigated by including asRNA in standard differential expression analyses and screening for significant interactions between asRNA/mRNA pairs as an additional layer of analysis on top of the existing experimental design.

**Future guidelines.** It is of great interest to determine the biological or technical processes driving asRNA and RNA relationships in metatranscriptomic data (3, 5, 7). In this study, we reviewed and summarized the characteristics of asRNA in metatranscriptomes from complex microbial communities. We have provided simple statistics of prevalence and quantities and elucidated the landscape of asRNA in complex communities. We found that genomic DNA contamination increases the level of asRNA, opening up the possibility of assessing the level of DNA contamination and, hence, data quality using asRNA. Conversely, if DNA sequencing data are available in parallel with the metatranscriptome, they could also be used as a means of adjusting asRNA levels to mitigate false positives. This study provides a methodological foundation for addressing asRNA in metatranscriptome analysis in general. As we have shown, asRNA expression does not seem to be caused by spurious promoters and may vary substantially between communities on multiple parameters such as functional categories, sequence diversity, and prevalence. If this variability is not acknowledged, important causal relationships could be overlooked, and incorrect interpretation of data could be performed. For example, transcription of asRNA may constitute a significant percentage of the data and may be associated with only a few genes. As sequencing data are inherently compositional, there will be an overrepresentation of spurious negative correlations with the remaining gene population, which cannot be amended using traditional quantitative data analysis (41). This is true regardless of whether the highly expressed genes are systematically related to the experiment or not. Such issues will be present only in communities where the fraction of asRNA constitutes a large portion of all the data, which we have shown to be highly variable. The naive solution will be to quantify mRNA only, ensuring that there are sufficient data for proper mRNA quantification. However, compositional data analysis methods exist, for which these issues can be amended (41, 42). As a minimum, we encourage paying attention to highly expressed genes with high fractions of asRNA, e.g., >90%, and either naively discarding them from downstream analysis or performing a thorough investigation to verify their credibility using existing tools for detecting spurious open reading frames, such as AntiFam (35). Cutoff levels of asRNA will be arbitrary and should be determined in conjunction with careful assessment of the data before downstream analysis.

This study highlights that very little is known about asRNA expression in complex communities. Hypothesis-driven studies are warranted to investigate the function and characteristics of asRNAs in metatranscriptomes. Integrating asRNA analysis with existing methodologies and pipelines, such as GWASs and differential expression analyses, will facilitate clarification of its role in cell processes and regulation (9). We believe that asRNA in complex communities is an important part of the metatranscriptome machinery, and interesting discoveries may lie ahead, driven by the approaches highlighted in this study.

mSystems®

## MATERIALS AND METHODS

A total of 56 strand-specific metatranscriptomes from anaerobic digester (43), water (23), permafrost soil (32), and human gut (4) environments were considered and reanalyzed in this study. Metatranscriptomic reads were mapped to metagenome-assembled genomes (MAGs) assembled from single samples or reference genomes provided by the respective studies (see Table S1 in the supplemental material).

**Anaerobic digester.** Anaerobic digester samples originated from a laboratory-scale short-chain-fatty-acid stimulation experiment. Slurry taken directly from an anaerobic digester at the Fredericia wastewater treatment plant (55.552301, 9.721908) was incubated in reactors and subjected to a spike-in of acetate. Two samples for DNA sequencing on the Nanopore and Illumina platforms were taken immediately before the spike-in (raw sequencing data available in the Sequence Read Archive [SRA] under accession number PRJNA603870). Library preparation for Illumina sequencing was described previously (43). Library preparation for Nanopore sequencing was carried out using the SQK-LSK108 kit (Oxford Nanopore Technologies, UK) according to the manufacturer's protocol, without the optional shearing step. The DNA libraries were sequenced using FLO-MIN106 MinION flow cells (Oxford Nanopore Technologies, UK) on a MinION mk1b system (Oxford Nanopore Technologies, UK). The raw fast5 data were base called using Guppy v2.2.2 with flipflop settings (Oxford Nanopore Technologies, UK). The Nanopore reads were filtered, removing short reads (<4,000 bp), and the adapters were then removed using Porechop v0.2.3 (44) and filtered based on quality using filtlong v0.1.1 (https://github.com/rrwick/Filtlong). The quality-filtered reads were assembled using CANU v1.8 (45), circular contigs were aligned to themselves using nucmer v3.1 (46), and the overlap was trimmed manually. The assembly was polished with Nanopore reads using minimap2 v2.14 (47) and Racon v1.3.2 (48), followed by polishing using medaka v0.4.3 twice (Oxford Nanopore Technologies, UK), and finally polished with minimap2 v2.14 and Racon v1.3.2 using Illumina data reported previously (43). The reads were then mapped back to the assembly using minimap2 v2.14 (47) to generate coverage depth files to assist automatic binning using metabat2 v2.12.1 (49). The generated MAGs were quality assessed using checkM v1.0.11 (50) with the "lineage_wf" option. Gene detection and annotation were performed using prokka v1.13 (51) with standard settings apart from "–metagenome –kingdom Bacteria." Sampling for RNA sequencing was done immediately before and at regular intervals up until 6 h after the spike-in (raw sequencing data available in the SRA under accession number PRJNA603157). Experimental details and protocols for RNA sequencing were described previously (43).

**Water.** Water samples originated from a previous study by Beier and colleagues (23), who analyzed functional redundancy in freshwater, brackish water, and saltwater communities. DNA sequencing data were downloaded from the SRA under accession number PRJEB14197. As some samples were sequenced multiple times, we included only the sequencing run with the most reads for each sample. Sequence reads were trimmed for adapters using cutadapt v1.16 (52) and assembled using megahit v1.1.3 (53). The reads were then mapped back to the assembly using minimap2 v2.12 (47) to generate coverage depth files to assist automatic binning using metabat2 v2.12.1 (49). A total of 147 MAGs were generated and quality assessed using checkM v1.0.11 (50) with the lineage_wf option. Gene detection and annotation were performed using prokka v1.13 (51) with standard settings apart from "–kingdom Bacteria –metagenome." RNA sequencing data were downloaded from the SRA under accession number PRJEB14197 and trimmed using cutadapt v1.16 (52).

**Bog and fen.** Bog and fen samples originated from a study by Woodcroft and colleagues (32), who analyzed carbon processing in thawing permafrost soil communities. We chose to analyze only metatranscriptomes from bog and fen samples due to insufficient sample sizes in the other groups. We obtained the raw sense and antisense gene expression data by personal communication with the authors of that study, which were mapped and quantified as described previously (32). We also received the gene sequences for annotation. Assembly and quality statistics of all 1,529 MAGs generated in this study are available in Data Set S3 from that study.

**Human gut.** Human gut samples originated from a study by Bao and colleagues (4), who analyzed antisense transcription in human gut communities. We downloaded sense and antisense gene expression data from Data Sheet 2 from Bao et al. (4). We focused only on the 47 species reported in Data Sheet 1 from Bao et al. (4), whose curated genomes were downloaded from an archived version of the FTP NCBI database (https://ftp.ncbi.nlm.nih.gov/genomes/archive/old_refseq/Bacteria). The genomes were quality assessed using checkM v1.0.11 (50) with the lineage_wf option.

**Mapping and quantifying RNAseq data.** Trimmed RNA sequencing data from the anaerobic digester and water samples were mapped using bowtie2 v2.3.4.1 (54) to their respective metagenomes with settings "-a –very-sensitive –no-unal." Downstream quantification was done using the python script BamToCount_SAS.py (https://github.com/TYMichaelsen/Publications/blob/master/BamToCount_SAS.py), parsing the bam files using pysam v0.15.0 (https://github.com/pysam-developers/pysam) and gff files using Biopython v1.72 (55).

**Filtering and preprocessing of expression data.** After the acquisition of data from all four studies, we retained only samples with a total read count of >1 million reads. Each data set was then filtered separately with the following steps: (i) include only protein-coding genes, (ii) remove genes with zero counts in >75% of samples to filter out rarely expressed genes, and (iii) remove observations with total read counts of <10 to filter low-expressed genes with a low signal-to-noise ratio. Transcripts per million (TPM) (56) were calculated from these data sets. Next, a binomial test for significant antisense transcription was done as described previously (4). Specifically, artifacts introduced by cDNA synthesis and amplification are known problems for antisense RNA detection (8), which leads to false-positive detection of antisense transcription. Let $q$ be the probability of having antisense reads when in reality there are none (i.e., the inherent technical error/false-positive rate). We then have a null hypothesis of observing

*q* or lower (i.e., one-tailed test) and test if the true percentage of antisense reads deviates from this. If the observed *P* value, after correction using the Benjamini-Hochberg procedure (57), is less than 0.05, we consider that the gene has antisense transcription. We used a *q* value of 0.01, as suggested previously (4), but also tested a *q* value of 0.05 as a more conservative estimate for comparison. From here, two data sets were generated for each community: one fulfilling previous criteria (sample-wide data set) and another specifically to analyze genome-wide antisense expression (genome-wide data set). The genome-wide data set was subjected to an additional filtering step. First, all genomes/MAGs failing to reach the criteria for a medium-quality MAG (completeness of ≥50% and contamination of <10%), determined by the Genomic Standards Consortium (58), were removed. Second, all genomes with <50 genes above the detection limit (total read count of ≥10) were removed for each sample separately. This was done to ensure that enough genes were above the detection limit to accurately calculate genome-wide estimates.

**Enrichment analysis.** We used the Clusters of Orthologous Groups (COG) database (59) as the basis for enrichment analysis. COG annotations were generated using the online version of eggNOG v1 (60) with standard settings. To test for the enrichment/reduction of a specific COG category in a set of genes, *n*, drawn from the entire population of genes, *N*, a hypergeometric test was performed. We calculated the probability of having *k* genes associated with the given COG in set *n*, given that there were *K* genes in the remaining population, $N - n$. If the *P* value for a one-tailed test for over- or underrepresentation of the COG category was below 0.05 after Benjamini-Hochberg correction (57), we considered it to be enriched or reduced, respectively. Two-tailed tests were performed by calculating the *P* values for both tails, taking the lowest value and multiplying it by 2 (61).

**DNA contamination experiment.** Nucleic acid was isolated from three samples originating from the same anaerobic digester as the one described above, using the RNeasy PowerMicrobiome kit (Qiagen) with a few adjustments. As the sample input, 500 μl wastewater was centrifuged at 14,000 × *g* for 10 min (4°C), and the supernatant was removed, allowing resuspension of the pellet in PM1 solution. Furthermore, the bead-beating step was performed with a FastPrep-24 instrument with four 40-s cycles at 6 m/s, and the samples were cooled on ice for 2 min between cycles. Purified nucleic acid from the three samples was split into two technical replicates, resulting in a total of six samples. All samples were DNase treated with the DNase Max kit (Qiagen). Prior to rRNA depletion, the six samples were diluted to 500 ng RNA but with a spike-in of ~130 ng DNA (from their original samples, respectively) in half of the samples. Ribosomal depletion was carried out with the Ribo-Zero (bacteria) magnetic kit (Illumina) according to the manufacturer's recommendations but using half the specified volumes of reagents for all reaction mixtures. The rRNA-depleted samples were prepped for RNAseq with the TruSeq stranded total RNA sample prep kit (Illumina) according to the manufacturer's recommendations but using half the specified volumes of reagents for all reaction mixtures. The final RNAseq library was sequenced on a HiSeq 2500 instrument (Illumina) with HiSeq Rapid SR cluster kit v2 (Illumina) and HiSeq Rapid SBS kit v2 (50 cycles; Illumina). Additional scan and incorporation mixes were added to support the sequencing of additional cycles.

**Data availability.** The raw sequencing data are available in the SRA under accession number PRJNA603157.

## SUPPLEMENTAL MATERIAL

Supplemental material is available online only.

**FIG S1**, TIF file, 1.8 MB.
**FIG S2**, TIF file, 1.9 MB.
**FIG S3**, TIF file, 1.7 MB.
**TABLE S1**, DOCX file, 0.01 MB.

## ACKNOWLEDGMENTS

We thank Joel A. Boyd for providing the raw gene expression and annotation data from the study by Woodcroft and colleagues (32) as well as personal communication about study details. We thank Erika Yashiro for performing the Illumina sequencing of anaerobic digester samples.

T.Y.M. and M.A. were supported by the Villum Foundation. N.S. was supported by a European Research Council grant (ERC MetaPG-716575).

T.Y.M., J.B., R.H.K., and M.A. are also employed by DNASense ApS. The other authors declare no conflict of interest.

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
