## [Reviewer comments · mSystems]

The signal and the noise – characteristics of antisense RNA in complex microbial communities

Thomas Michaelsen, Jakob Brandt, Caitlin Singleton, Rasmus Kirkegaard, Johanna Wiesinger, Nicola Segata, and Mads Albertsen

Corresponding Author(s): Thomas Michaelsen, Aalborg University

Review Timeline:

Submission Date:	September 12, 2019
Editorial Decision:	October 19, 2019
Revision Received:	December 16, 2019
Accepted:	January 4, 2020

Editor: Athanasios Typas

Reviewer(s): The reviewers have opted to remain anonymous.

Transaction Report:

DOI: <https://doi.org/10.1128/mSystems.00587-19>

Mr. Thomas Yssing Michaelsen
Aalborg University
Center for Microbial Communities
Fredrik Bajers Vej 7H
Aalborg 9200
Denmark

Re: mSystems00587-19 (The signal and the noise - characteristics of antisense RNA in complex microbial communities)

Dear Mr. Thomas Michaelsen:

your minireview has now been reviewed by 2 experts, who have provided some constructive feedback on points that would need to be clarified, e.g. defining what is regarded as "transcriptional noise", being more concise on how transcriptional noise links to technical, biological or random (genetic) effects, and at the end explaining better the rationale behind the claim that asRNA is majorly not due to transcriptional noise. You can find the comments at the end of this letter.

Your minireview is likely to be accepted once answers are provided to reviewers and the indicated changes are made. If you would like a brief biographical sketch of each author (limit, 150 words) to be published at the end of your article, please submit text and photos with your modified manuscript. Please refer to the instructions posted at <http://www.asm.org/images/journals/Files/AuBiosITA.pdf>.

We now offer our authors the services of ASM's contracted artist, Patrick Lane of ScEYence Studios. This art enhancement service is free of charge to authors of minireviews and full-length reviews, and turnaround time is fast. Please contact Patrick on receiving this letter. Complete contact information for Patrick and further instructions are posted at <http://www.asm.org/images/journals/Files/ArtEnhancementITA.pdf>.

Please return your modified manuscript within 60 days; if you cannot complete the modification within this time period, please contact me. If you decide that you do not want to modify the manuscript and wish to submit it to another journal, please notify me of your decision immediately so that the manuscript can be formally withdrawn.

To submit the modified manuscript, log onto the eJP submission site at <https://msystems.msubmit.net/cgi-bin/main.plex>. If you cannot remember your password, click the "Can't remember your password?" link and follow the instructions on the screen. Go to Author Tasks and click the appropriate manuscript title to begin the resubmission process. The information you entered when you first submitted the paper will be displayed. Please update the information as necessary. Provide (1) point-by-point responses to the issues raised by the reviewers as file type "Response to Reviewers," not in your cover letter, and (2) a PDF file that indicates the changes from the original submission (by highlighting or underlining the changes) as file type "Marked Up Manuscript - For Review Only."

To avoid unnecessary delay in publication should your modified manuscript be accepted, it is important that you submit your entire manuscript digitally and that all elements meet the technical

requirements for production. Be sure that your submission contains the entire manuscript, not just the items that have been modified. Before you submit your modified manuscript, I strongly recommend that you check your digital images by running them through Rapid Inspector, an automated figure preflighting tool available at the following URL:

<http://rapidinspector.cadmus.com/RapidInspector/zmw/>

Thank you for submitting your minireview to mSystems.

Sincerely,
Nassos Typas
Editor, mSystems

Journals Department
Fax: 1-202-942-9355

Reviewer comments:

Reviewer #1 (Comments for the Author):

In this manuscript, Michaelsen and colleagues perform a meta-analysis of antisense mapping reads in metatranscriptomic data sets. They briefly introduce antisense RNA (asRNAs), then perform a reanalysis of four diverse metatranscriptomic datasets including an anaerobic digester, water, bog and fen environments, and the human gut. They show that in most environments with high levels of antisense read counts, these mostly originate from a few discrete loci. They additionally show that most gene sets annotated by COG are depleted in genes with greater than 95% of reads mapping to the antisense direction. Finally, while in the abstract they state "we tested the hypothesis that antisense transcription is primarily the product of transcriptional noise and found mixed support, suggesting that the total observed antisense RNA in complex communities arises from a compounded effect of both random, biological and technical factors", they make a much stronger statement in the section title stating "Levels of asRNA are genome-specific and not driven by transcriptional noise."

The first major comment I have on the manuscript is regarding this final point - the inference that antisense RNAs are not majorly the product of transcriptional noise. The evidence supporting this claim is unclear to me. In the relevant section of the manuscript, the authors first test whether the number of genes with significant antisense transcription is better explained by sample, or genome composition, showing that genome composition has a larger effect. They then reproduce the analysis of Llorens-Rico et al., showing that promoter motifs occurrence more or less follows an exponential distribution dependent on AT content, consistent with combinatorics and suggesting no strong selection for or against these motifs at the scale of the genome. Both of these observations are consistent with the majority of antisense promoters being the product of random mutation with no strong effects on organismal fitness.

The final analysis the authors perform in this section (figure 2D) is to compare total antisense reads

counts to genome content. The authors find that antisense read count controlled for genome size has no clear relationship with genome AT content. While this is not clearly stated, the authors seem to take this as evidence that antisense reads are not the product of transcriptional noise.

It is unclear to me how this lack of a relationship provides much evidence one way or the other for antisense reads being the product of transcriptional noise. Why should we expect Figure 2D to show any particular relationship between total antisense read count and AT content under the null hypothesis that most antisense transcripts are non-functional? My naive expectation is that the total antisense read count would be dominated by genome abundance in the sample, and metabolic/transcriptional activity of that strain, plus maybe an allowance for the author's finding that the majority of antisense reads come from a very small number of loci (though it is unclear if this extends to per genome read counts). Since these factors haven't been corrected for, it's difficult to understand how any residual underlying relationship could be seen - though even then it's not really clear what the null hypothesis being tested might be.

In light of this, I would suggest the authors either clearly explain their reasoning in this section, and the conclusions taken from it, or re-evaluate their approach to testing the transcriptional noise hypothesis.

A second major point is regarding the interpretation of antisense transcription: it seems that the authors assume that genes with ~100% antisense reads are genuine antisense transcripts. Given the (to me anyway) unexpected enrichment of these in their data (fig 1B), is it not equally likely these are the product of annotation errors and represent either bona fide intergenic non-coding RNAs, or coding RNAs on the opposite strand? The following paper provides some illustrations of how false gene annotations have propagated in protein databases as the result of annotation errors that may explain (some of) this observation - at the very least this should be considered, though proposals for how to deal with it would also be very welcome:

Eberhardt, R. Y. et al. AntiFam: a tool to help identify spurious ORFs in protein annotation. Database 2012, bas003 (2012).

Höps, W., Jeffryes, M. & Bateman, A. Gene Unprediction with Spurio: A tool to identify spurious protein sequences. F1000Res. 7, 261 (2018).

Finally, the authors' recommendations for handling antisense data in metatranscriptomic studies could be made more clear - quoting their abstract, I am not sure how antisense reads provide insight into "technical details about data quality" or "novel insight into the biology of complex microbial communities". Is this referring to the section of text on page 7 lines 280 to 290? I would suggest clarifying what their future guidelines for antisense transcription, and perhaps moderating some of the claims in the abstract.

Minor points:

1. It would be useful to define what the authors mean by transcriptional noise early in the manuscript, as this term is used for several related but distinct concepts in different subfields. The following reference may be helpful in this regard:

Raser, J. M. & O'Shea, E. K. Noise in gene expression: origins, consequences, and control. Science 309, 2010-2013 (2005).

2. It would also be useful to present the additional evidence that is available suggesting that most

antisense transcripts are not well conserved, in addition to the study from Llorens-Rico:

Raghavan, R., Sloan, D. B. & Ochman, H. Antisense transcription is pervasive but rarely conserved in enteric bacteria. *MBio* 3, (2012).

Shao, W., Price, M. N., Deutschbauer, A. M., Romine, M. F. & Arkin, A. P. Conservation of transcription start sites within genes across a bacterial genus. *MBio* 5, e01398-14 (2014).

3. It would also be useful to provide some additional cases where antisense transcript has clearly been shown to be functional. This review includes a number of good examples and mechanisms:

Sesto, N., Wurtzel, O., Archambaud, C., Sorek, R. & Cossart, P. The excludon: a new concept in bacterial antisense RNA-mediated gene regulation. *Nat. Rev. Microbiol.* 11, 75-82 (2013).

4. It seems that the reference list is incomplete - for instance on page 2 line 53 I was unable to find a reference for Förstner et al 2016.

5. On page 6, lines 189 - 195, could the authors provide some more details about the proteins mentioned, what genomes they were associated with, and also clarify what they mean by finding "no spurious amplifications of primer regions when inspecting the coverage profile"?

6. In figure 2A, would this result change using say, a homology based annotation like EggNOG (<http://eggnogdb.embl.de>)? This might increase the number of genes with COG annotations. In addition, while the text says "the prevalence of genes without COG annotation were significantly enriched", this does not seem to be reflected in the figure.

7. Could the authors please describe the binomial test on page 6, line 239 in the methods? It's a bit unclear at the moment what this is testing for.

Reviewer #2 (Comments for the Author):

The mini-review from Thomas Michaelsen et al., summarizes the knowledge about asRNA expression in complex communities, as well as the current approaches to handle strand specific metatranscriptomic RNAseq data. In order to strengthen their points regarding the importance of integrating the strand specific signal into the analysis they performed an explorative meta-analysis of the quantitative and functional properties of asRNA in different datasets. Based on their analysis the authors concluded that including asRNA in standard differential expression analysis would facilitate clarification of its role in cell processes and regulation.

The mini review is very well written from a highly experienced team in the field. The metatranscriptomics field is still in its infancy therefore such contributions that can serve as guidelines for bioinformaticians and experimentalists are very welcomed. A couple of comments more because I am obliged as reviewer to be critical would be:

- The authors decided to look into 5 different datasets and compare the levels of asRNA. Four out of the five datasets have 7-9 samples, which is quite a small number to make generalizations about the % of antisense reads in each environment. I would find more interesting to examine only one dataset and see how the biological conclusions may change following the three approaches,

removing asRNA, integrating them, ignoring them. Nevertheless, I understand that maybe the authors do not want for a mini review to spend more time on that, its just that as a reader I would be much more inconvenienced for the value of asRNA in microbial communities if indeed the results or biological conclusions change.

- The authors found that in anaerobic digester, bog and fen samples 2-3 genes accounted for up to 90% of all asRNA communities. How is the situation in the other 2 datasets? Especially if in humans the situation is similar, considering that the total percentage of asRNA is just 1% do you still think that we lose a lot ignoring them?

- In Line 275-280 the authors discuss the value of high through put omics for building metabolic networks and suggest that asRNA expression should be integrated in such analysis. How could asRNA be integrated for building metabolic networks and in metabolic network analysis?

- In Line 281 the authors claim that metatranscriptomics studies are often explorative. Metatranscriptomics studies are still very expensive so I am not sure what the authors mean that they are explorative. Could they add some references in this sentence to clarify what is "explorative" in their opinion and what hypothesis driven?

Dear Nasos Typas,

We appreciate very much that you and the reviewers have taken the time to review our manuscript and find our manuscript of high quality and relevant for publication in the mSystems Journal (in a revised form). We appreciate very much all their comments and suggestions, which we have implemented in the revised manuscript. We have provided comments to the points raised by the reviewers in red.

Kind regards,
Thomas Y. Michaelsen and Mads Albertsen

Reviewer #1 (Comments for the Author):

In this manuscript, Michaelsen and colleagues perform a meta-analysis of antisense mapping reads in metatranscriptomic data sets. They briefly introduce antisense RNA (asRNAs), then perform a reanalysis of four diverse metatranscriptomic datasets including an anaerobic digester, water, bog and fen environments, and the human gut. They show that in most environments with high levels of antisense read counts, these mostly originate from a few discrete loci. They additionally show that most gene sets annotated by COG are depleted in genes with greater than 95% of reads mapping to the antisense direction. Finally, while in the abstract they state "we tested the hypothesis that antisense transcription is primarily the product of transcriptional noise and found mixed support, suggesting that the total observed antisense RNA in complex communities arises from a compounded effect of both random, biological and technical factors", they make a much stronger statement in the section title stating "Levels of asRNA are genome-specific and not driven by transcriptional noise."

Major comments

Comment 1:

The first major comment I have on the manuscript is regarding this final point - the inference that antisense RNAs are not majorly the product of transcriptional noise. The evidence supporting this claim is unclear to me. In the relevant section of the manuscript, the authors first test whether the number of genes with significant antisense transcription is better explained by sample, or genome composition, showing that genome composition has a larger effect. They then reproduce the analysis of Llorens-Rico et al., showing that promoter motifs occurrence more or less follows an exponential distribution dependent on AT content, consistent with combinatorics and suggesting no strong selection for or against these motifs at the scale of the genome. Both of these observations are consistent with the majority of antisense promoters being the product of random mutation with no strong effects on organismal fitness.

We agree with the reviewer that the formulation is unclear and might cause confusion and have tried to make it more clear in the manuscript.

1) The ANOVA analysis was based on antisense expression, parameterized using a statistical test for significant antisense expression, and summarised for each genome as the

fraction of genes with significant antisense expression. We did not test genome composition, but average fraction of antisense RNA between genomes. Our findings suggest that variability in fraction of antisense RNA between genomes is larger than the sample-to-sample variability (line 247-251). This is still true after adjusting for total RNA read count and AT content (line 251-257).

2) This analysis was decoupled from the analysis of promotor-motif occurrence, where antisense expression was parameterized as number of antisense reads per Mbp genome. We have made it more clear that these two parametrizations are complementary (see line 283-286).

We agree with the reviewer that antisense promotor-motifs are primarily the product of random mutations, but fail to see how this should restrict antisense expression itself to be random. Albeit promotor-motifs are present, there exists reasons why they may not be leading to transcription (Wade and Grainger 2014). This is a relevant addition to our reasoning, which we have added at line 289-291.

The final analysis the authors perform in this section (figure 2D) is to compare total antisense reads counts to genome content. The authors find that antisense read count controlled for genome size has no clear relationship with genome AT content. While this is not clearly stated, the authors seem to take this as evidence that antisense reads are not the product of transcriptional noise.

We agree with the reviewer. The lack of a relationship between antisense expression and promotor-motif occurrence as evidence against transcriptional noise is too strong a statement and have rephrased to a less strong statement (line 289-291).

It is unclear to me how this lack of a relationship provides much evidence one way or the other for antisense reads being the product of transcriptional noise. Why should we expect Figure 2D to show any particular relationship between total antisense read count and AT content under the null hypothesis that most antisense transcripts are non-functional? My naive expectation is that the total antisense read count would be dominated by genome abundance in the sample, and metabolic/transcriptional activity of that strain, plus maybe an allowance for the author's finding that the majority of antisense reads come from a very small number of loci (though it is unclear if this extends to per genome read counts). Since these factors haven't been corrected for, it's difficult to understand how any residual underlying relationship could be seen - though even then it's not really clear what the null hypothesis being tested might be. In light of this, I would suggest the authors either clearly explain their reasoning in this section, and the conclusions taken from it, or re-evaluate their approach to testing the transcriptional noise hypothesis.

We agree that more detail is needed to explain the association between antisense read count and AT content. The "naive expectations" mentioned by the reviewer were very constructive. Inspired by this we have performed simple univariate linear regression of antisense read counts on potential confounders such as transcriptional activity and genome

coverage (line 278-280). We find limited explanatory power of these and our main point that of all *known* explanatory variables the genome still explains most of the variance still holds.

We agree with reviewer that there are too many unknown factors to have an established null-hypothesis of the causal factors controlling total antisense read count. We have refrained from using this terminology in the context of transcriptional noise (see for example the abstract).

Comment 2:

A second major point is regarding the interpretation of antisense transcription: it seems that the authors assume that genes with ~100% antisense reads are genuine antisense transcripts. Given the (to me anyway) unexpected enrichment of these in their data (fig 1B), is it not equally likely these are the product of annotation errors and represent either bona fide intergenic non-coding RNAs, or coding RNAs on the opposite strand? The following paper provides some illustrations of how false gene annotations have propagated in protein databases as the result of annotation errors that may explain (some of) this observation - at the very least this should be considered, though proposals for how to deal with it would also be very welcome:

Eberhardt, R. Y. et al. AntiFam: a tool to help identify spurious ORFs in protein annotation. Database 2012, bas003 (2012).

Höps, W., Jeffryes, M. & Bateman, A. Gene Unprediction with Spurio: A tool to identify spurious protein sequences. F1000Res. 7, 261 (2018).

This is a very important point raised by the reviewer and we concur that genes with almost exclusively antisense reads have a high risk of being annotation errors. In the manuscript we referred to the study by (Bao et al. 2015), in which they highlight the ambiguity of genes with high fractions of antisense reads and assume that these are likely non-coding RNA (Bao et al. 2015). To further emphasize this we have added an additional comment on line 208-209. We also tested the software proposed by the reviewer and did not detect any spurious annotations using AntiFam (Eberhardt et al. 2012). We attempted to implement the spurio software (Höps, Jeffryes, and Bateman 2018), but we were not able to scale it for 100,000s of protein sequences as there is no parallelization available for the software. Furthermore, it writes to the same output folder (this cannot be changed), so any attempt to parallelize will result in overwriting existing files. We deeply regret this and hope the reviewer acknowledges that fixing this problem will take too much of an effort relative to the improvement of the paper, as we already have tested the AntiFam software.

We agree that proposals to handle this are needed and at line 343-346 in the manuscript we suggest to flag genes with e.g. the fraction of antisense reads above a certain threshold, e.g. >90%, either naively removing them before normalization and analysis or conducting a thorough verification using existing tools as mentioned by the reviewer.

Comment 3:

Finally, the authors' recommendations for handling antisense data in metatranscriptomic studies could be made more clear - quoting their abstract, I am not sure how antisense reads provide insight into "technical details about data quality" or "novel insight into the biology of complex microbial communities". Is this referring to the section of text on page 7 lines 280 to 290? I would suggest clarifying what their future guidelines for antisense transcription, and perhaps moderating some of the claims in the abstract.

To backup our claim that antisense reads may provide insight into technical details about data quality we have performed an experiment with artificially induced DNA contamination using spike-in (line 219-225). We found that genes with high DNA coverage also had increased levels of antisense RNA. This opens the possibility of both assessing the level of DNA contamination in the sample, as well as using DNA coverage as a way of correcting antisense levels (line 327-331).

The reviewer is correct that we highlight how antisense reads may provide novel insight into the biology of complex microbial communities on page 7 line 280-290 in the previous manuscript. We have rephrased this to make it more clear according to the reviewer recommendations (line 308-309).

Finally, we have also moderated our claims in the abstract.

Minor comments

1. It would be useful to define what the authors mean by transcriptional noise early in the manuscript, as this term is used for several related but distinct concepts in different subfields. The following reference may be helpful in this regard:

Raser, J. M. & O'Shea, E. K. Noise in gene expression: origins, consequences, and control. *Science* 309, 2010-2013 (2005).

We have defined transcriptional noise early on in the manuscript (line 38-43) and added the reference suggested by the reviewer.

2. It would also be useful to present the additional evidence that available suggesting that most antisense transcripts are not well conserved, in addition to the study from Llorens-Rico:

Raghavan, R., Sloan, D. B. & Ochman, H. Antisense transcription is pervasive but rarely conserved in enteric bacteria. *MBio* 3, (2012).

Shao, W., Price, M. N., Deutschbauer, A. M., Romine, M. F. & Arkin, A. P. Conservation of transcription start sites within genes across a bacterial genus. *MBio* 5, e01398-14 (2014).

These studies are interesting and a good addition to the manuscript. We have added them on line 52-54.

3. It would also be useful to provide some additional cases where antisense transcript has clearly been shown to be functional. This review includes a number of good examples and mechanisms:

Sesto, N., Wurtzel, O., Archambaud, C., Sorek, R. & Cossart, P. The excludon: a new concept in bacterial antisense RNA-mediated gene regulation. *Nat. Rev. Microbiol.* 11, 75-82 (2013).

This is a valuable reference, which we already use. We have added a sentence to the introduction (line 54).

4. It seems that the reference list is incomplete - for instance on page 2 line 53 I was unable to find a reference for Förstner et al 2016.

This was an error, the correct reference has been added to the text and reference list.

5. On page 6, lines 189 - 195, could the authors provide some more details about the proteins mentioned, what genomes they were associated with, and also clarify what they mean by finding "no spurious amplifications of primer regions when inspecting the coverage profile"?

Encouraged by the reviewers comment on verifying the credibility of annotation we went back and revised the blast hits behind our claims pointed out by the reviewer. After updating our criteria for alignment (query coverage $\geq 25\%$, identity $\geq 70\%$, and e-value ≤ 0.001), we were not able to obtain any hits. We believe that this disqualifies our claims for the mentioned proteins and we have therefore amended the sentence to reflect that on line 192-198.

By the sentence "no spurious amplification of primer regions when inspecting the coverage profile" we mean that any spuriously high gene expression can be caused by areas of the gene have high primer affinity, causing a spike in the coverage profile that ultimately leads to over-estimating the expression of the gene. This has now been clarified on line 199-200.

6. In figure 2A, would this result change using say, a homology based annotation like EggNOG (<http://eggnogdb.embl.de>)? This might increase the number of genes with COG annotations. In addition, while the text says "the prevalence of genes without COG annotation were significantly enriched", this does not seem to be reflected in the figure.

On request of the reviewer we reran the COG annotation through the online available version of eggNOG v1 (<http://eggnogdb.embl.de/#/app/emapper>) using standard settings. As suspected by the reviewer, we managed to annotate substantially more genes (line 230). Overall, the patterns remained the same with the exception of Mobilome: prophages,

transposons (category X), which was significantly enriched in the gut and anaerobic digester communities, while no changes were observed for the remaining communities (line 231-233). As eggNOG is based on non-supervised clustering and phylogeny of an extensive database, we have implemented this tool instead of prokka (Seemann 2014) for functional annotation and re-written the methods and results accordingly.

The significant enrichment of genes without annotation is shown in the bottom of figure 2, as the horizontal red bars. These indicate an enrichment relative to the background population of genes, which across all five communities were significant as indicated by " <0.001 " to the right of each bar. To avoid confusion we have added "Note the break between 10% and 50% on the x-axis, where the group of non-annotated extends across the whole figure from left to right" to the figure legend.

7. Could the authors please describe the binomial test on page 6, line 239 in the methods? It's a bit unclear at the moment what this is testing for.

This is described in the methods line 133-140: "Next, a binomial test for significant antisense transcription was done as described in (Bao et al. 2015). Specifically, artifacts introduced by cDNA synthesis and amplification are known problems for antisense RNA detection (Thomason and Storz 2010), which leads to false positive detection of antisense transcription. Let q be the probability of having antisense reads when in reality there is none (i.e. the inherent technical error/false positive rate). We then have a null hypothesis of observing q or less (i.e. a one-tailed test) and test if the true percentage of antisense reads deviates from this. If the observed p-value, after correction using the Benjamini-Hochberg procedure (Benjamini and Hochberg 1995), is less than 0.05 we consider that the gene has antisense transcription. We used a q value of 0.01 as suggested by (Bao et al. 2015), but tested also $q = 0.05$ as a more conservative estimate for comparison."

We have simplified the sentence, which we believe make it more clear, and added a back-reference to the methods at line 244.

Reviewer #2 (Comments for the Author):

The mini-review from Thomas Michaelsen et al., summarizes the knowledge about asRNA expression in complex communities, as well as the current approaches to handle strand specific metatranscriptomic RNAseq data. In order to strengthen their points regarding the importance of integrating the strand specific signal into the analysis they performed an explorative meta-analysis of the quantitative and functional properties of asRNA in different datasets. Based on their analysis the authors concluded that including asRNA in standard differential expression analysis would facilitate clarification of its role in cell processes and regulation.

The mini review is very well written from a highly experienced team in the field. The metatranscriptomics field is still in its infancy therefore such contributions that can serve as guidelines for bioinformaticians and experimentalists are very welcomed. A couple of comments more because I am obliged as reviewer to be critical would be:

Comments

1. The authors decided to look into 5 different datasets and compare the levels of asRNA. Four out of the five datasets have 7-9 samples, which is quite small number to make generalizations about the % of anti sense reads in each environment. I would find more interesting to examine only one dataset and see how the biological conclusions may change following the three approaches, removing asRNA, integrating them, ignoring them. Nevertheless, I understand that maybe the authors do not want for a mini review to spend more time on that, its just that as a reader I would be much more inconvenienced for the value of asRNA in microbial communities if indeed the results or biological conclusions change.

We agree with the reviewer that a deep-dive into the how the biological conclusions would change dependent on including antisense reads or not is very interesting and of great relevance setting an example for future studies, we believe this is out of scope of this minireview. This is also mentioned by the reviewer.

2. The authors found that in anaerobic digester, bog and fen samples 2-3 genes accounted for up to 90% of all asRNA communities. How is the situation in the other 2 datasets? Especially if in humans the situation is similar, considering that the total percentage of asRNA is just 1% do you still think that we lose a lot ignoring them?

We have added numbers for the other datasets at line 194-195. We agree that the importance of antisense RNA is dependent on the relative abundance compared to the total metatranscriptome. We have added a comment to reflect that on line 340-341.

3. In Line 275-280 the authors discuss the value of high throughput omics for building metabolic networks and suggest that asRNA expression should be integrated in such analysis. How could asRNA be integrated for building metabolic networks and in metabolic network analysis?

We think the most interesting approach would be to infer suppression based on the ratio of antisense to sense reads - if this ratio is high that would imply the gene is suppressed. This would add an additional third change of state for a gene, besides the trivial up- and down-regulation. We have added this reasoning on line 308-309

4. In Line 281 the authors claim that metatranscriptomics studies are often explorative. Metatranscriptomics studies are still very expensive so I am not sure what the authors mean that they are explorative. Could they add some references in this sentence to clarify what is "explorative" in their opinion and what hypothesis driven?

We agree with the reviewer that the definition of the term "explorative" may be vague. We adopt the definition from (Franzosa et al. 2015), which is a hypothesis-searching and -generating study in which groups are compared in search of biomarkers that can differentiate groups. This has been clarified on line 312-313.

References

- Bao, Guanhui, Mingjie Wang, Thomas G. Doak, and Yuzhen Ye. 2015. "Strand-Specific Community RNA-Seq Reveals Prevalent and Dynamic Antisense Transcription in Human Gut Microbiota." *Frontiers in Microbiology* 6 (September): 896.
- Benjamini, Yoav, and Yosef Hochberg. 1995. "Controlling the False Discovery Rate: A Practical and Powerful Approach to Multiple Testing." *Journal of the Royal Statistical Society: Series B (Methodological)*. <https://doi.org/10.1111/j.2517-6161.1995.tb02031.x>.
- Eberhardt, Ruth Y., Daniel H. Haft, Marco Punta, Maria Martin, Claire O'Donovan, and Alex Bateman. 2012. "AntiFam: A Tool to Help Identify Spurious ORFs in Protein Annotation." *Database: The Journal of Biological Databases and Curation* 2012 (March): bas003.
- Franzosa, Eric A., Tiffany Hsu, Alexandra Sirota-Madi, Afrah Shafquat, Galeb Abu-Ali, Xochitl C. Morgan, and Curtis Huttenhower. 2015. "Sequencing and beyond: Integrating Molecular 'Omics' for Microbial Community Profiling." *Nature Reviews. Microbiology* 13 (6): 360–72.
- Höps, Wolfram, Matt Jeffryes, and Alex Bateman. 2018. "Gene Unprediction with Spurio: A Tool to Identify Spurious Protein Sequences." *F1000Research* 7 (March): 261.
- Seemann, Torsten. 2014. "Prokka: Rapid Prokaryotic Genome Annotation." *Bioinformatics* 30 (14): 2068–69.
- Thomason, Maureen Kiley, and Gisela Storz. 2010. "Bacterial Antisense RNAs: How Many Are There, and What Are They Doing?*", November. <https://doi.org/10.1146/annurev-genet-102209-163523>.
- Wade, Joseph T., and David C. Grainger. 2014. "Pervasive Transcription: Illuminating the Dark Matter of Bacterial Transcriptomes." *Nature Reviews. Microbiology* 12 (9): 647–53.

January 4, 2020

Mr. Thomas Yssing Michaelsen
Aalborg University
Center for Microbial Communities
Fredrik Bajers Vej 7H
Aalborg 9200
Denmark

Re: mSystems00587-19R1 (The signal and the noise - characteristics of antisense RNA in complex microbial communities)

Dear Mr. Thomas Yssing Michaelsen:

Both reviewers and myself are satisfied with the revised version of the manuscript, which I am happy to let you know has been accepted. I am forwarding it to the ASM Journals Department for publication. For your reference, ASM Journals' address is given below. Before it can be scheduled for publication, your manuscript will be checked by the mSystems production editor, Ellie Ghatineh, to make sure that all elements meet the technical requirements for publication. She will contact you if anything needs to be revised before copyediting and production can begin. Otherwise, you will be notified when your proofs are ready to be viewed.

Sincerely,

Athanasios Typas
Editor, mSystems

Journals Department
Supplemental figure 3: Accept

Supplemental figure 2: Accept

Supplemental figure 1: Accept

Reviewer comments:

Reviewer #1 (Comments for the Author):

Michaelsen and co. have responded thoughtfully to my previous comments. In particular their new regression analysis of antisense expression in metagenomic samples with consideration of various potential confounders should be an interesting starting point for further exploration of the origin and regulation of antisense expression. Between this, and their moderation of their previous claims, I feel the manuscript is much improved. I have no further comments.